# Multi-color photonic integrated circuits based on homogeneous integration of quantum cascade lasers

Dominik Burghart ✉, Kevin Zhang, Wolfhard Oberhausen, Anna Köninger, Gerhard Boehm & Mikhail A. Belkin ✉

We demonstrate an InP-based mid-infrared photonic integrated circuit processed from a wafer in which two distinct quantum cascade laser active regions are grown in different areas on the same InP crystal. A passive InGaAs waveguiding layer is epitaxially deposited on top of the entire InP substrate prior to the laser active region growth to optically couple the lasers emission and to multiplex their emission wavelengths to a single output waveguide. The method demonstrated in this work enables the creation of monolithic photonic integrated circuits with emission wavelength spanning the entire 3-15 μm spectral range and it is of interest for a wide range of applications.

The mid-infrared spectral region (mid-IR, λ≈3-15 μm) contains numerous narrow absorption lines associated with vibrational modes of different molecular groups. Broadband infrared spectrometers that rely on thermal sources are widely used to determine the chemical and structural composition of samples based on their mid-IR absorption "fingerprint". The development of quantum cascade lasers (QCLs) has revolutionized mid-IR spectroscopic instrumentation by offering compact, mass-producible, electrically-pumped laser sources with orders of magnitude narrower linewidth and higher brightness compared to thermal sources[1–4] and, more recently, with an ability to provide frequency-comb emission[5,6].

Room-temperature QCLs have been developed for operation in the entire mid-IR spectral range[1–3,7]. However, the gain bandwidth of an individual QCL is limited to only a relatively small portion of the mid-IR band. Since spectroscopic applications often desire to cover a wide frequency spectrum, broadening of the available gain bandwidth has been one of the key focuses of QCL research. QCL active regions with multiple lower and/or upper laser levels to broaden the available gain bandwidth were developed to that end[8–10]. Further increase of the QCL gain bandwidth was achieved by growing heterogeneous stacks of up to six different active regions in the laser waveguide core[11–13] which results, however, in deterioration of the laser performance due to gain-bandwidth constraints and due to the increase of thermal resistance with the number of laser stages[14,15]. Practical QCL systems for broadband spectroscopy, microscopy, or multi-species gas sensing use free-

space optical setups with beam splitters or movable mirrors to combine the outputs of multiple individual QCL chips[16–19].

Unlike diode lasers, QCL operation is based on intersubband transitions, and their performance is known not to be strongly affected by heterostructure defects, as evidenced, e.g., by QCL high-performance operation as metasurface lasers with cavities etched through the entire QCL heterostructure[20,21] or when grown on strongly lattice-mismatched substrates[22–24]. This property simplifies monolithic integration of dissimilar QCL active regions on the same chip based on selective epitaxial growth/regrowth methods. On-chip optical interconnects and wavelength multiplexers may then be used to build mid-IR photonic integrated circuits (PICs) combining dissimilar active elements on the same chip.

Here, we experimentally demonstrate the first mid-IR PICs with dissimilar QCL active regions monolithically integrated and optically coupled on the same InP crystal. In this approach, the entire PIC chip is a single crystal which results in its high reliability and robustness to environmental changes. This is particularly important for QCL-based PICs since QCLs have very high threshold power density of over 10 kW/cm² [25]. Additionally, monolithic integration of all devices on the same crystal simplifies mass production of PICs based on standard III-V semiconductor processing technology. The results presented in this paper open the possibility of developing chip-scale monolithic QCL sources with emission spanning the entire mid-IR spectral range and is of interest for broadband spectroscopy, multi-species gas sensing,

Walter Schottky Institute, Technical University of Munich, Am Coulombwall 4, Garching 85748, Germany. ✉e-mail: Dominik.Burghart@wsi.tum.de; Mikhail.Belkin@wsi.tum.de

multi-band free-space communications and other applications that require high-performance laser operation with emission wavelengths spread over large mid-IR bandwidth.

In the near-infrared spectral range, monolithic integration of III-V lasers with other photonic elements on an InP crystal based on multiple epitaxial growth steps[26] as well as heterogeneous integration of III-V lasers with other active elements on silicon-on-isolator (SOI) or other waveguiding platforms[27], are used to produce near-infrared PICs containing dissimilar active and passive optical elements. Both approaches result in commercially successful products[27,28]. However, established near-infrared waveguiding platforms are not transparent across the entire mid-IR spectral range, particularly beyond 5 μm[29]. Additionally, QCLs have threshold power densities ~10–100 times higher than that of diode lasers[7,25], which makes thermal management of heterogeneously-integrated QCLs more difficult than that of diode lasers, which is already a challenge in heterogeneous PICs[27]. At present, only low-duty-cycle short pulse QCL operation have been achieved for heterogeneously-integrated QCLs and all devices operated at wavelengths shorter than 5 μm[30–32]. In contrast, the InP-based waveguiding platform is transparent in 3-15 μm[33–35] and monolithic integration of QCLs on InP allows one to follow standard approaches for thermal management developed for InP-based QCLs.

The schematic of the mid-IR PIC concept presented in this work is shown in Fig. 1. We use an InP/InGaAs-based passive waveguiding platform that was recently experimentally demonstrated to possess very low mid-IR optical losses[33–35]. First, an InGaAs/InP passive waveguiding layer is grown on an entire n-doped InP wafer; then different QCL active regions can be selectively grown at particular areas of the wafer on top of the waveguiding layer, as shown in Fig. 1(a). The waveguides of the processed QCLs are tapered to couple light to the passive waveguiding layer, as shown schematically in Fig. 1(b). Finally, the outputs from the QCLs are multiplexed to a single waveguide, see Fig. 1b. This approach enables free scaling of the bandwidth of the PIC, limited only by the performance of individual QCL active regions.

## Results

### Selective quantum cascade laser active region growth on InP

While it is expected that intersubband devices are well-suited for selective growth/re-growth on a wafer, this has not yet been reported experimentally. As a starting point, we produced two nominally identical active regions grown on different parts of the wafer in two separate epitaxial runs. This approach allows us to directly compare the performance of the lasers processed from the originally-grown and regrown areas on the same wafer. We chose to grow QCLs with active regions containing only 15 quantum cascade laser stages following the demonstrations in refs. 15,36–39 that QCLs with a reduced number of stages (10–15) offer significantly lower thermal resistance compared to traditional mid-IR QCLs that have ~30–40 active region stages. We note that QCLs with a reduced number of stages typically operate at a factor of ~2-3 lower bias voltage compared to traditional mid-IR QCLs, but their threshold current density is typically ~2–3 times higher to compensate for lower mode overlap with the active region as discussed, e.g., in ref. 39.

The epilayers were grown by solid-source molecular beam epitaxy (MBE) on a 2-inch-diameter n-doped InP substrate. Details of the laser active regions, etch-stop layers, and waveguide structures are given in Methods. The growth started with the lower waveguide cladding layer, followed by an etch-stop layer. The first active region (AR1) was grown on the entire wafer, as shown in Fig. 2a. The active region was then removed from specific areas defined by optical lithography, down to the etch-stop layer, which was selectively removed afterwards, as shown in Fig. 2b. After photoresist removal, the wafer was cleaned (see Methods) and inserted back into the MBE reactor. The etch-stop layer followed by an identical active region (AR2) was grown on the wafer as shown in Fig. 2c. The second active region in the wafer areas with the

double-stacked active region was selectively removed using a pattern defined by optical lithography. The result is shown in Fig. 2d. After photoresist removal and wafer cleaning, the wafer was brought back to the MBE reactor for the third time and the upper waveguide cladding layers of InGaAs and InP were grown by MBE on the entire wafer as shown in Fig. 2e. The microscope image of the resultant wafer is shown in Fig. 2f. Active region sections were positioned in 0.5 mm-wide stripes. Defects can be seen at the interface between the grown/regrown QCL sections; however, the epilayer morphology appears normal for both grown and regrown sections away from these interfaces.

Ridge-waveguide lasers were processed from AR1 and AR2 sections of the wafer to compare the performance of the devices with the active region grown initially and the ones with the active region regrown. The laser ridges were positioned at the center of the sections, far away from defect areas. A comparison of the performance of the devices is given in Fig. 3. The results demonstrate that QCL active regions regrown on the wafer produce no significant deterioration in device performance. Small differences in the spectral position of light emission and dynamic range seen in Fig. 3 are within the range of typical MBE parameters drift.

### Photonic integration of distinct quantum cascade laser active regions

To demonstrate integration and multiplexing of QCLs with different active regions grown on the same semiconductor crystal, we have prepared a 2-inch-diameter InP wafer with a passive waveguiding layer made of lightly-n-doped ($3 \times 10^{15}$ cm$^{-3}$) 1.25 μm-thick InGaAs passive waveguide core surrounded by slightly doped InP cladding layers ($7 \times 10^{15}$ cm$^{-3}$). The detailed layer structure of our wafer is given in Methods. The thickness of the passive waveguide layer was chosen so that the effective index for TM$_{00}$ slab waveguide mode in the passive waveguide is smaller than the effective index for TM$_{00}$ QCL optical mode for our 7- to 8 μm-wide ridge-waveguide QCLs[25]. Tapering of the QCL waveguide ridge to below ~4 μm width is then used to couple light from the QCL active region to the passive waveguide layer[25]. Lightly doped passive waveguide structure, combined with n-doped InP substrate allowed for current extraction from the QCL active regions into the device substrate[25].

A QCL active region (AR) designed for a gain peak at 6.25 μm (AR 1) and at 7.3 μm (AR 2) were grown in selective areas of this wafer following the approach shown in Fig. 2. The AR center wavelengths were chosen to target absorption lines of NO$_2$ and SO$_2$ trace gases. The scanning electron microscope (SEM) image of the cleaved side of the resulting wafer structure is shown in Fig. 4a. The image allows us to estimate the width of the defect region at the interface between the grown/regrown QCL sections to be ~10 μm, indicating that very high integration density of dissimilar devices can be attained with this approach in principle. Note that the passive waveguiding layer is unaffected by the regrowth process, providing a path to connect sections of the wafer with different QCL active regions.

To verify the performance of the two active regions, the PIC wafer shown in Fig. 4a was first processed into Fabry-Perot ridge-waveguide lasers with the laser ridges etched through both the active region and passive waveguide layer sections. Figure 4b, c show the typical performance of 10 μm-wide 5 mm-long ridge waveguide lasers processed from the wafer sections with AR 1 and AR 2, respectively.

For the demonstration of the PIC based on the wafer shown in Fig. 4a, we processed QCLs with AR1 and AR2 into distributed feedback (DFB) lasers using sidewall corrugation of the laser ridge waveguides[40,41]. The DFB gratings are designed for laser operation at the wavelengths of 6.25 μm (AR1) and 7.29 μm (AR2), corresponding to absorption lines of NO$_2$ and SO$_2$ gases, respectively. We tapered the QCL waveguides widths to couple light to the passive waveguide layer as in ref. 25. A schematic of the DFB QCL ridge with the taper and the

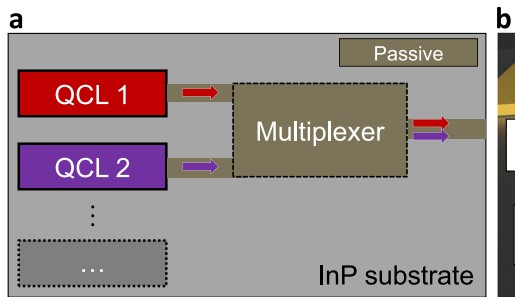

**Fig. 1 | Photonic integrated circuit concept for quantum cascade laser bandwidth multiplexing. a** General schematic of the broadband mid-infrared photonic integrated circuit concept based on selective growth of distinct quantum cascade laser active regions. **b** Detailed schematic of the mid-infrared photonic integrated circuit reported in this work.

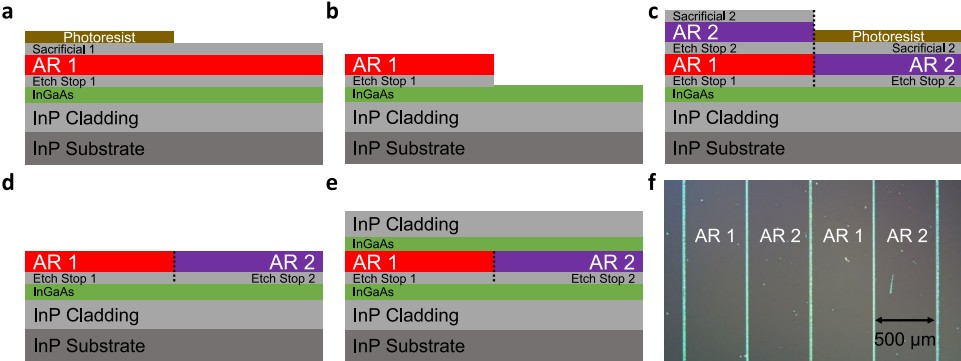

**Fig. 2 | Integration of quantum cascade laser active regions on the same InP crystal. a–e** Processing steps for integrating different active region stacks on a single InP crystal using molecular beam epitaxy. See Methods for further details. **f** Microscope image of the resultant wafer. Lines of defects along the interfaces between grown and regrown quantum cascade laser active regions can be seen.

simulation results of the beam coupling from the QCL into the passive waveguide layer are shown in Fig. 5a. Over 80% coupling efficiency of the power of the forward-propagating QCL mode into the passive waveguide is predicted by simulations at both 6.25 μm and 7.3 μm wavelengths. This corresponds to the coupler insertion loss of less than -1 dB at both wavelengths. Details of the DFB grating and the taper design are given in Methods. The ridge waveguides for DFB QCLs and the tapers are etched simultaneously with inductively-coupled plasma reactive ion etching (ICP-RIE) after definition of a silicon nitride (SiN) hard mask with electron beam lithography (EBL). Both AR1 and AR2 active regions were grown with a similar thickness in order to allow simultaneous stopping within the 750 nm thick common lower cladding layer (see Methods for the layer structure of the wafer).

Finally, all passive components, including both the passive waveguides and the multiplexer (cf. Fig. 1b) are patterned with EBL, transferred to a thin SiN hard mask and etched with ICP-RIE. The schematic and the simulated performance of the multiplexer is shown in Fig. 5b and Fig. 5c, d, respectively. The multiplexer performance was confirmed experimentally by processing a similar multiplexer from a wafer with only passive waveguide layers and measuring transmission through ports 1 and 2 at λ ≈ 6.25 μm and 7.4 μm, respectively, using external pumping. Based on the results shown in in Fig. 5d, we estimate the insertion losses of the multiplexer at -0.5 dB and -1 dB for 6.2-6.3 μm and 7.3-7.4 μm wavelengths, respectively. We further note that losses for the passive waveguides are estimated to be in the range 1–3 dB/cm for 6.3–7.4 μm wavelengths based on the values measured earlier for a similar waveguiding system[34].

Following the definition of the passive photonic components, the QCL processing is finalized, including deposition and opening of the SiN isolation layer on the laser ridges, metal contacts definition, and electroplating. After substrate thinning to about 150 μm thickness and backside metal deposition, two-color PIC chips are cleaved and mounted on copper blocks in an episide up configuration. The optical microscope image of a two-color PIC chip is shown in Fig. 6a. The SEM image of one of the QCL ridge section near the beginning of the taper is shown in Fig. 6b. The approximate position of this cross section in the device is indicated in Fig. 6a. The SEM images of the multiplexer sections are shown in Figs. 6c, d with the corresponding positions in the PIC indicated in Fig. 6a. Finally, Fig. 6e shows the SEM image of the laser ridge from the cleaved back facet of the laser.

## Performance of the two-color photonic integrated circuit

The results of the PIC testing are shown in Fig. 7. Figure 7a shows the current-voltage and the light-output-current characteristics of the two laser sources. The optical power for both DFB lasers was collected from the facet of the common output waveguide of the PIC (left side of the PIC shown in Fig. 6a). We note that the methods that we chose for DFB fabrication resulted in reduction of the laser performance compared to the performance to Fabry-Perot devices reported in Fig. 4. The reason for that is the choice of the narrower ridge width for the DFB QCL fabrication, compared to that used for fabrication of Fabry-Perot devices. In order to provide sufficiently strong DFB coupling strength by a relatively small sidewall corrugation, the laser ridge widths were reduced to the 7–8 μm range (see Methods for DFB QCL laser structure details). This reduction of the ridge width combined with the unexpected passive waveguide layer thickness being 6.5% higher than specified in the growth sheet resulted in the partial leakage of the laser mode into the passive waveguide layer, which led to the increase in devices' threshold current density. Evidence for higher-than-expected passive layer thickness was found during device

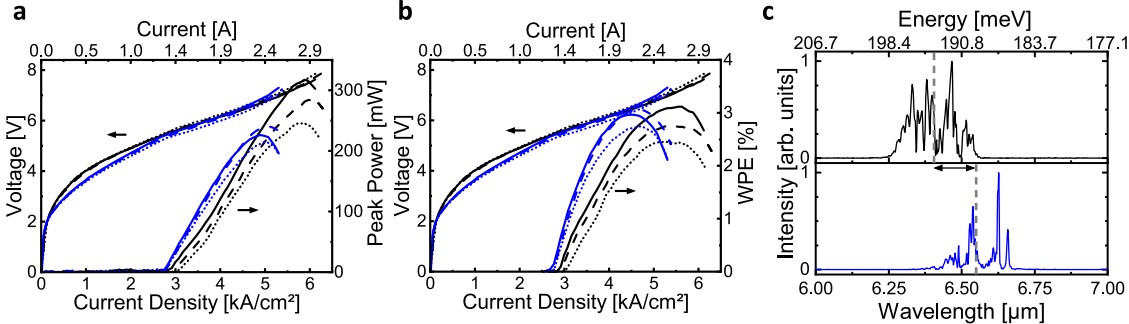

**Fig. 3 | Comparison of the devices' performance processed from grown and regrown sections of the wafer. a, b** Light-output-current-voltage characteristics (**a**) and single-facet wall-plug-efficiency (**b**) for three ridge-waveguide lasers processed from the originally-grown active region (black solid, black dashed, and black dotted lines) and from the nominally identical active region regrown in selective areas of the same wafer (blue solid, blue dashed, and blue dotted lines). **c** Typical emission spectra of the quantum cascade lasers shown in (**a, b**) for the case of the devices with originally-grown active region (upper panel) and the regrown active region (lower panel). All the tested devices are 4-mm-long 12 μm wide ridge waveguide Fabry-Perot lasers. Devices were operated in pulsed mode at room temperature with 300 ns current pulses at 10 kHz pulse repetition frequency.

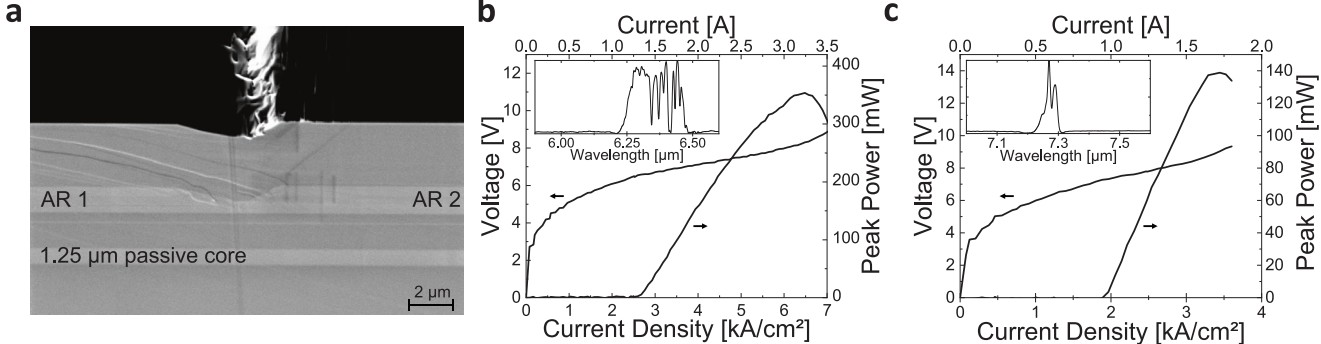

**Fig. 4 | Details of the photonic integrated circuit wafer with two distinct quantum cascade laser active regions. a** Scanning electron microscopy image of a cleaved facet of a grown photonic integrated circuit wafer showing the passive waveguide layer and the two active region sections. Growth defects at the interface between the two active region sections could be seen. However, the passive waveguide layer has no such defects. **b, c** Performance of ridge-waveguide Fabry-Perot quantum cascade lasers processed from active region (AR) 1 (**b**) and AR 2 (**c**) sections of the wafer shown in (**a**). Devices were operated in pulsed mode at room temperature. Insets show the emission spectra of the lasers.

fabrication and was later confirmed with SEM images. Analysis of the difference in the modal overlap with the active region for the 7.5 μm-wide DFB and 10 μm-wide Fabry-Perot devices is given in the Supplementary Information Section 1. Some fine tuning of the DFB widths and resulting PIC design would allow to avoid this problem in the future. Nevertheless, pulsed room temperature operation was achieved for both devices at the target DFB wavelengths in individual and simultaneous operation as shown in Fig. 7b.

Figure 7c shows the profile of the far-field emission of the two-color PIC for the AR1-laser and AR2-laser operation. The light output from the common PIC output waveguide facet was collimated by an anti-reflection-coated aspherical molded black diamond lens with a numerical aperture of 0.85. The beam profiles were measured by a microbolometric-camera-based WinCamD-IR-BB beam profiler at a distance of 50 cm from the PIC. The two beam profiles were obtained for a fixed alignment of the optical setup by turning on either the first or the second laser on the PIC. The resulting beam intensities for both colors show nearly-circular far-field profiles for both wavelengths with excellent overlap, demonstrating benefits of the photonic integration approach in achieving nearly-perfect modal overlap of distinct QCLs.

Due to the relatively high threshold current density of DFB QCLs compared to their wider-ridge-width Fabry-Perot analogs, CW operation of only one of the two DFB QCLs in the PIC (at λ = 6.25 μm) was attainable at room temperature. The results are shown in Fig. 7d. The

etalon effect of the passive waveguide section on the DFB QCL laser performance can clearly be observed as regular oscillations in the light-current-voltage characteristics displayed in Fig. 7d. The period of optical power oscillation in the L-I device characteristic is 0.9 W of electrical power dissipation and it is consistent with the tuning rate of the 6.25 μm DFB lasers of -0.74 cm$^{-1}$ per 1 W of electrical power dissipation and the Fabry-Perot mode spacing of the 2.4 mm-long passive waveguide section of the PIC of $\Delta\nu = 1/(2n_{eff}L) \approx 0.67$ cm$^{-1}$, where $n_{eff} \approx 3.13$ is the effective refractive index of TM$_{00}$ mode in the passive waveguide. This power oscillation can be suppressed by introducing a nearly-perfect anti-reflection coating on the output facet as shown in the Supplementary Information Section 2.

In conclusion, we demonstrated the first mid-IR PIC that monolithically integrate two dissimilar QCL active region on the same InP crystal and optically couple the output of the two lasers to a single output facet using passive optical waveguides and wavelength multiplexers. We presented a PIC configuration in which a common passive waveguide layer can be used to connect selectively-grown QCL active regions, while avoiding defects at the interfaces between QCL active regions. We confirmed that the performance of QCL active regions is virtually unaffected by regrowth on selected InP crystal areas. We successfully demonstrated room-temperature operation of two-color mid-IR PICs that combine DFB QCLs processed from two distinct QCL active regions and a wavelength multiplexer processed in the passive

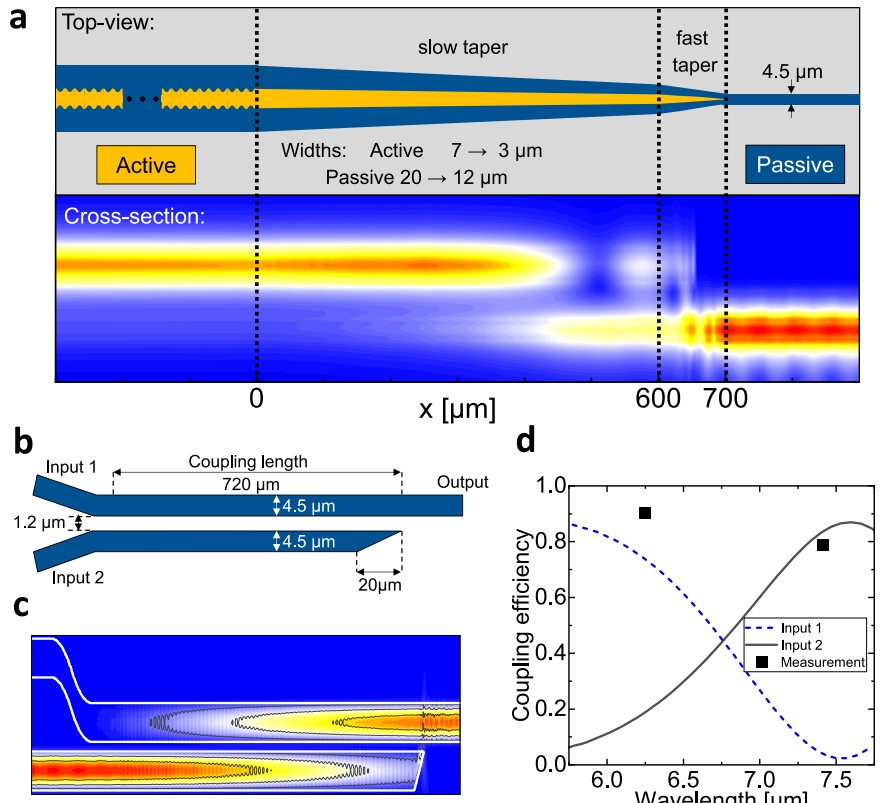

**Fig. 5 | Schematic and simulations of the two-color photonic integrated circuit.**
**a** Schematic of the distributed feedback quantum cascade laser coupled to the passive waveguide (top panel) and simulation results of the optical intensity transfer from the quantum cascade laser to the passive waveguide at $\lambda = 7.3\,\mu m$ (bottom panel). A quantum cascade laser waveguide taper transfers the optical power from the active region into the passive waveguide core. **b** Schematic of the wavelength multiplexer. **c** Simulation of the power transfer for the fundamental transverse magnetic mode from the input 2 waveguide to the output waveguide at $\lambda = 7.3\,\mu m$ (cf. panel (**b**)). **d** Calculated power transmission for the fundamental transverse magnetic mode from inputs 1 and 2 to the output as a function of wavelength in the multiplexers shown in (**b**, **c**). Squares show experimental measurements.

waveguide layer. Far-field profile measurements of the processed PIC demonstrate nearly perfect modal overlaps of the two emitters.

The approach presented in this work enables the development of chip-scale monolithic QCL sources with emission spanning the entire mid-IR spectral range and is of interest for broadband spectroscopy, multi-species gas sensing, multi-band free-space communications and other applications that require high-performance laser operation with emission wavelengths spread over large mid-IR bandwidth.

## Methods

### Details of the wafer structure used for the active region regrowth testing

Three growth steps are required to complete the wafer characterized in Fig. 3. The structure was grown by MBE on an InP substrate n-doped to $3\text{-}6 \times 10^{18}\,cm^{-3}$. The initial growth starts with our standard buffer of 100 nm of $Ga_{0.47}In_{0.53}As$ layer n-doped to $2.5 \times 10^{18}\,cm^{-3}$ followed by a 4 μm-thick InP lower cladding layer n-doped to $2 \times 10^{16}\,cm^{-3}$. These layers are immediately followed by a 500 nm-thick $Ga_{0.47}In_{0.53}As$ separate confinement heterostructure (SCH) layer n-doped to $2 \times 10^{16}\,cm^{-3}$, a 50 nm-thick InP etch stop layer n-doped to $2 \times 10^{16}\,cm^{-3}$, another 50 nm-thick $Ga_{0.47}In_{0.53}As$ layer n-doped to $2 \times 10^{16}\,cm^{-3}$, as well as the first of the two nominally identical 800 nm-thick active region stacks consisting of 15 QCL stages, capped with a 50 nm-thick layer of $Ga_{0.47}In_{0.53}As$ n-doped to $2 \times 10^{16}\,cm^{-3}$ and a sacrificial 50 nm-thick InP layer n-doped to $2 \times 10^{16}\,cm^{-3}$. The layer thicknesses (in Angstroms) for a single period of the active region heterostructure, starting from the injection barrier, are **28**/*17*/**10**/22/*30*/**11**/21/*29*/**13**/20/**20**/**17**/19/*17*/**16**/17/*15*/**16**/16/*13*/**18**/*28*/**21**/26/**25**/*24*. The underlined layers are n-doped to

$3.15 \times 10^{17}\,cm^{-3}$, $Al_{0.62}In_{0.38}As$ barriers are listed in bold, $Ga_{0.35}In_{0.65}As$ wells are listed in cursive and $Ga_{0.47}In_{0.53}As$ wells listed with regular letters. Following this first growth, a photolithographic mask is defined to allow removal of the active region from parts of the wafer. At first the InP sacrificial layer and the active region are selectively removed by hydrochloric acid $HCl:H_2O$ [1:1] and phosphoric acid $H_3PO_4:H_2O_2:H_2O$ [1:1:2], respectively. After the removal of the resist mask by acetone and cleaning the wafer with $O_2$ plasma in a barrel asher, the top 50 nm of InP are selectively removed by $HCl:H_2O$ [1:1]. This additional wet chemical cleaning step is enabled by the implementation of the before-mentioned sacrificial InP layer during the first growth. This step helps to avoid the possible influence of carbon pollutants from the photo-resist during the next MBE growth step.

The second epitaxy step starts with another 50 nm-thick InP etch stop layer n-doped to $1.5 \times 10^{16}\,cm^{-3}$, followed by a 50 nm-thick $Ga_{0.47}In_{0.53}As$ layer n-doped to $2 \times 10^{16}\,cm^{-3}$, the exact same active region heterostructure, an $Ga_{0.47}In_{0.53}As$ layer n-doped to $2 \times 10^{16}\,cm^{-3}$ and a 50 nm-thick sacrificial InP layer n-doped to $1.5 \times 10^{16}\,cm^{-3}$. The same hydrochloric and phosphoric etch chemistry is then used to remove the second active region stack from the areas on top of the initially grown active region. Then the same cleaning steps are performed to prepare the wafer for a final growth of a 500 nm-thick $Ga_{0.47}In_{0.53}As$ upper SCH layer n-doped $2 \times 10^{16}\,cm^{-3}$, a 4.5 μm-thick InP upper cladding layer n doped to $1.5 \times 10^{16}\,cm^{-3}$, and a 500 nm-thick InP plasmon confinement and contact layer n-doped to $5 \times 10^{18}\,cm^{-3}$. This results in a wafer with two nominally identical active regions in different sections of the wafer grown in two separate epitaxial runs.

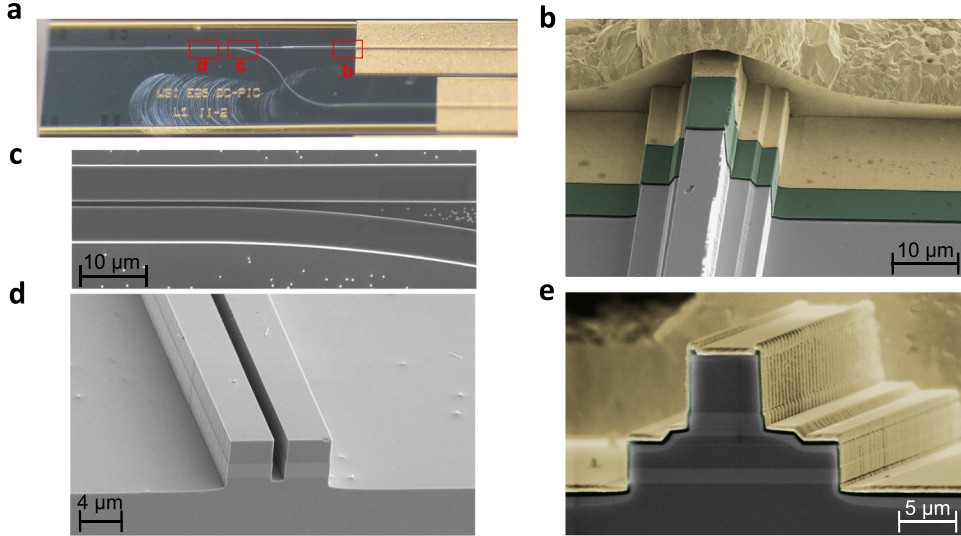

**Fig. 6 | Processed two-color photonic integrated circuit. a** Microscope image of the two-color photonic integrated circuit. Approximate positions of the images shown in the (**c**, **d**) of this figure are indicated in red with the letters corresponding to that of the panels. **b** Scanning electron microscopy image of the onset of the laser ridge taper section. **c**, **d** Scanning electron microscopy images of the multi-plexer section of the photonic integrated circuit. **e** Scanning electron microscopy image of the back laser facet.

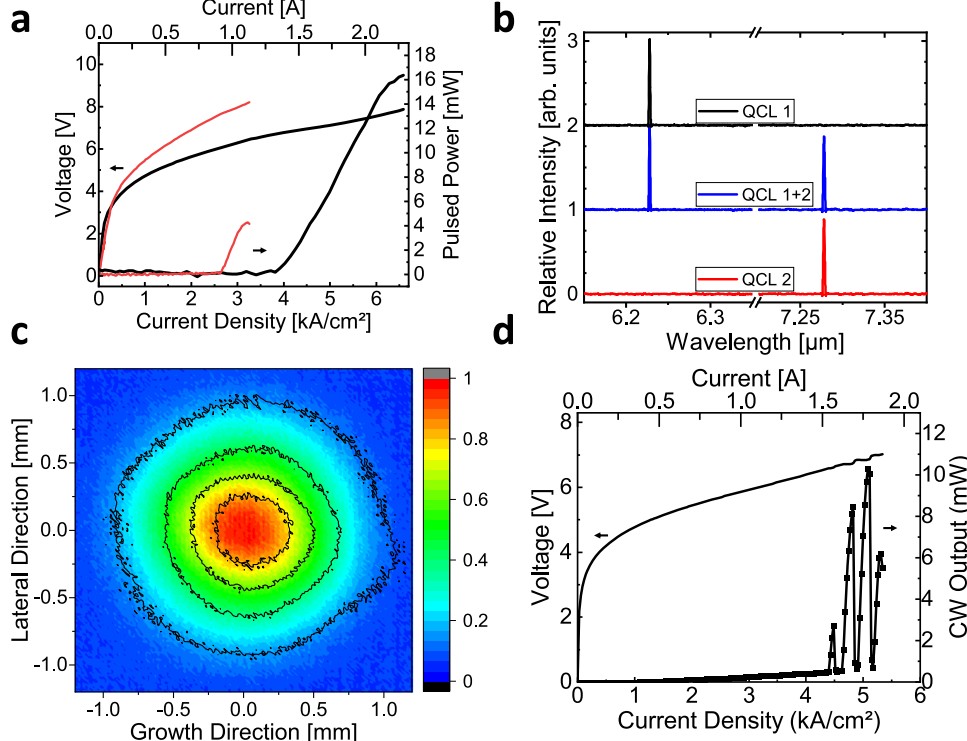

**Fig. 7 | Characterization of the dual-color photonic integrated circuit. a** Light-output-current-voltage characteristics of distributed feedback quantum cascade laser 1 targeting 6.25 μm (thick black lines) and distributed feedback quantum cascade laser 2 targeting 7.29 μm (thin red lines). The optical power is collected from a single passive output waveguide of the photonic integrated circuit. **b** Mid-infrared spectra for both lasers in the photonic integrated circuit shown in (**a**) operated individually (top and bottom panels) as well as simultaneously (middle panel). **c** Color contour plot of beam profiles for emission at 6.25 μm (black contours) and 7.29 μm (colored intensity profile) wavelength at 50 cm distance from the photonic integrated circuit described in (**a**). The measurements for the data shown in (**a**–**c**) were performed at room temperature with the photonic integrated circuit biased with 70–250 ns pulses at 150–350 kHz repetition frequencies. **d** Light-output-current-voltage characteristics for the quantum cascade laser 1 of a photonic integrated circuit operated in continuous-wave at room temperature. The optical power is collected from the passive output waveguide.

## Details of the wafer structure used for fabrication of the PIC devices

The structure was grown by MBE on an InP substrate n-doped to $3–5 \times 10^{18}$ cm$^{-3}$. The growth started with a lower passive waveguide cladding layer made of a 2 μm-thick InP layer n-doped to $2 \times 10^{17}$ cm$^{-3}$, a 1.5 μm-thick InP layer n-doped to $5 \times 10^{16}$ cm$^{-3}$, and a 1.5 μm-thick InP cladding layer n-doped to $7 \times 10^{15}$ cm$^{-3}$. These layers were followed by the passive waveguide core made of a 1.25 μm-thick $Ga_{0.47}In_{0.53}As$ layer

n-doped to $3 \times 10^{15}$ cm$^{-3}$. Slight doping of the passive waveguide core and cladding layers was introduced to enable current injection into the QCLs grown on top of the passive waveguide layer through the substrate. Next, a 1.5 μm-thick upper InP cladding layer n-doped to $7 \times 10^{15}$ cm$^{-3}$ was grown, followed by a 500 nm- thick InP upper cladding layer n-doped to $5 \times 10^{16}$ cm$^{-3}$, and a 50 nm-thick etch-stop $Ga_{0.47}In_{0.53}As$ layer n-doped to $8 \times 10^{16}$ cm$^{-3}$. This layer could also be used for lateral current injection using a lateral contact, although this function was not utilized in our particular PIC realization.

Two distinct QCL active regions were then grown on top of the passive waveguide layer employing the same selective etch/regrowth techniques as described above. Both layer stacks share a 750 nm-thick lower InP cladding layer n-doped to $2 \times 10^{16}$ cm$^{-3}$, as well as a 5 μm-thick upper InP cladding with doping gradient from $2 \times 10^{16}$ cm$^{-3}$ next to the active region to $5 \times 10^{18}$ cm$^{-3}$ near the top metal contact. The core of the 6.25 μm QCL is made of a 600 nm-thick active region made of 11 quantum cascade stages with the layer structure described above surrounded by SCH layers of 700 nm-thick $Ga_{0.47}In_{0.53}As$ n-doped to $2 \times 10^{16}$ cm$^{-3}$ above and below the active region. The core of the 7.3 μm QCL is made of 600 nm-thick active region made of 15 quantum cascade stages described below surrounded by SCH layers of 800 nm-thick $Ga_{0.47}In_{0.53}As$ n-doped to $2 \times 10^{16}$ cm$^{-3}$ above and below the active region. The 6.25 μm QCL structure is grown first and the 7.3 μm QCL structure is grown second on the wafer areas in which with 6.25 μm active region is selectively removed following the procedure described in the previous Methods section. The layer thicknesses for a single the 7.3 μm QCL active region stage, in Angstroms, starting from the injection barrier is given as **31**/21/8/**58**/*10*/**40**/13/*38*/**12**/*31*/13/*27*/**15**/*25*/**19**/*25* with underlined layers doped $1 \times 10^{17}$ cm$^{-3}$, $Al_{0.62}In_{0.38}As$ barriers shown in bold, $Ga_{0.35}In_{0.65}As$ wells shown in cursive, and $Ga_{0.47}In_{0.53}As$ wells denoted with a regular font.

### Taper design

The thicknesses of the passive waveguide core and the QCL waveguide core layers, as well as the InP cladding layers of the PIC wafer were designed to have nearly the same taper configuration for transferring optical mode from the laser waveguide core to the passive waveguide. The taper design for the 7.3 μm QCL is shown in Fig. 5a. For the 6.25 μm QCL, a similar taper structure was used with the only difference is that the slow taper section was 200 μm longer. The fast taper section (cf. Fig. 5a) was identical for both lasers. The purposed of the fast taper section is to suppress reflections of light back to the QCLs from the taper end.

### Sidewall corrugated DFBs

DFB gratings for single-mode operation are produced by sinusoidal modulation of the laser ridge width, with an amplitude of ~350 nm on each side of the laser ridge. For the target wavelength of 6.25 μm, a ridge width of 7 μm was used, while an 8 μm ridge width was used for the 7.3 μm target wavelength. We estimate the coupling coefficient κ for such a grating to be about 10 cm$^{-1}$ for both target wavelengths. Grating periods were chosen to be 0.97 μm and 1.14 μm for the target wavelengths 6.25 μm and 7.3 μm respectively. All gratings have a quarter wavelength shift at the center of the laser ridge for improved longitudinal mode selection.

### PIC fabrication

The QCL cavities as well as the active outline of the taper structure for the active to passive coupler were etched simultaneously with reactive ion etching after definition of a hard mask with electron beam lithography. Both active stacks were grown with similar thickness in order to allow simultaneous stopping within the 750 nm-thick common lower InP cladding layer of QCLs (see the PIC wafer structure description above). An additional hard mask is structured to protect all active components while the 750 nm-thick

common InP lower cladding layer is wet-chemically removed prior to the dry chemical etching definition of the PIC passive components to reveal an epitaxially flat surface defined by the 50 nm-thick $Ga_{0.47}In_{0.53}As$ etch-stop layer grown on top of the passive waveguide InP upper cladding layer (see the PIC wafer structure description above). This planarization step is necessary to enable the use of electron beam lithography with thin resists for definition of the passive PIC components, including the multiplexer. Finally, the structure of all passive components is patterned with an e-beam, the e-beam pattern is transferred to a thin SiN hard mask, and the passive waveguide structures are dry-etched using a $CH_4/H_2$ plasma in an ICP-RIE. After the passive photonic structures are defined, the QCL process−including electrical insulation, contact definition, electroplating, and thinning to about 150 μm−is completed, and the samples are mounted in an episide-up configuration on a copper submount using Indium solder.

## Data availability

All data supporting the findings of this study are available within the main article and the Supplementary Information file. The raw data are available from the corresponding authors upon request.

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

## Acknowledgements
This work is supported by funding from the European Union's Horizon 2020 research and innovation program under grant agreement No. 101016956 PASSEPARTOUT (to M.A.B.), in the context of the Photonics Public Private Partnership and by the German Research Foundation (DFG) under Grant No. 463411319. (to M.A.B.).

## Author contributions
M.A.B. conceived the project, with K.Z. and D.B. providing support. The structure design was carried out by D.B., with contributions from W.O. and K.Z. All structures were fabricated and characterized by D.B. Wafer growth and related guidance was provided by G.B., with assistance from A.K. All authors discussed the results. D.B. and M.A.B. wrote the manuscript, and M.A.B. supervised the work.

## Funding

## Competing interests
The authors declare no competing interests.
