## [Transparent Peer Review file · Nature Communications]

Multi-Color Photonic Integrated Circuits Based on Homogeneous Integration of Quantum Cascade Lasers

Corresponding Author: Mr Dominik Burghart

Version 0:

Reviewer comments:

Reviewer #1

(Remarks to the Author)
see attached

Reviewer #2

(Remarks to the Author)
I co-reviewed this manuscript with one of the reviewers who provided the listed reports. This is part of the Nature Communications initiative to facilitate training in peer review and to provide appropriate recognition for Early Career Researchers who co-review manuscripts.

Reviewer #3

(Remarks to the Author)
In their paper, the authors demonstrate the growth of two quantum cascade lasers with different active regions and operation wavelengths and their integration with a passive waveguide on a common Indium Phosphide substrate. The results are very interesting as they provide a possible path for very broadband integrated mid-infrared sources in an integrated optics format. The results are very convincing and well supported by data and the paper should be published as is.

Version 1:

Reviewer comments:

Reviewer #1

(Remarks to the Author)
Most comments have been thoroughly addressed, and I recommend the paper for publication.

The authors have addressed some comments from our previous review. While for some comments, the authors have responded to them without any revision to the work.

1. In comment 6, it is advised to characterize the system with two lasers pumped simultaneously. It could be right that there are no constraints for operating both lasers simultaneously, while it is still necessary to validate the multi-color PIC based on this method. Please provide the spectrum at the output.

2. In comment 4, it is advised to give a comparison of hybrid/heterogeneous integration versus the monolithic approach in terms of efficiency, cost, scalability, and complexity. It is true that the monolithic chip is reliable and robust to environmental changes. While it remains to be discussed whether it can simplify mass production in III-V PICs. First, multiple MBE growth and lithography processes will significantly increase the difficulty of wafer preparation. Second, before we deposit the AR2 layer, half of the AR1 layer was etched away, which will increase the cost. Therefore, a discussion on efficiency and cost will be beneficial.

3. The whole paper is focused on the fabrication process, there is still significant room for improvement in the device and application. In figure 7 (d), why the LI curve oscillates greatly when J is larger than 4.5 KA/cm². The article provides two possible solutions without implementation. Will the device fulfill its designated function in this scenario? I do not intend to imply that this article lacks novelty. However, further improvement in the device's performance would enhance the article's persuasiveness.

Reviewer #2

(Remarks to the Author)

Version 2:

Reviewer comments:

Reviewer #1

(Remarks to the Author)

Most comments have been thoroughly addressed. And there's only one more comment:

The power oscillations were suppressed by applying an anti-reflection coating to the output waveguide facet, however, at the cost of requiring cooling (at 200°K) to achieve CW operation. Is this a common issue in DFB QCLs? How could this be improved in the future?

Reviewer #2

(Remarks to the Author)

Dear Reviewers,

We want to express our sincere thanks for carefully reading the manuscript and suggesting improvements. Reviewer 3 had no questions or comments and suggested the manuscript for publication as is. All the comments addressed below are from Reviewers 1 and 2, who jointly reviewed the manuscript and provided detailed feedback. Please find below our point-by-point response to your questions and comments. We use black font for the initial comment and blue font for our replies. Additionally, we are submitting an updated manuscript file in which your comments have been addressed. The modified portions of text in the revised manuscript are underlined.

Reviewer #1 and #2 (Comments to the Author):

Comment 1:

The threshold current density of the devices is relatively high. How does this compare with state-of-the-art QCLs in the literature?

Response: Thank you for making us aware that this key point is not addressed in enough detail.

In this work we use a relatively new configuration of a QCL active region, with a stage count reduced from 30-40 in typical devices to only 10-15. This approach simplifies operating devices continuous-wave when processed as ridge-waveguide lasers, without buried heterostructure processing. This approach was pioneered in Refs. ^{15,29-32} cited in the manuscript. QCLs with reduced stage count typically have lower voltage, compared to 'traditional' QCLs but higher current density to compensate for a smaller overlap integral of the optical mode with the active region. The benefit of this approach is that the thermal resistance of the active region is significantly reduced with thickness, further details are provided in Refs. ^{15,29-32}. Room-temperature threshold current density of 2-3kA/cm² of our devices is consistent with the state-of-the-art for QCLs with reduced stage count, see, e.g., Ref. ¹⁵.

In order to make this point clearer to the readers, we have added an underlined sentence in the first part of the result section:

We chose to grow QCLs with active regions containing only 15 quantum cascade laser stages following the demonstrations in Refs. ^{15,29-32} that QCLs with a reduced number of stages (10-15) offer significantly lower thermal resistance compared to traditional mid-IR QCLs that have ~30-40 active region stages. We note that QCLs with a reduced number of stages typically operate at a factor of ~2-3 lower bias voltage compared to traditional mid-IR QCLs, but their threshold current density is typically ~2-3 times higher to compensate for lower mode overlap with the active region as discussed, e.g., in Ref. ³².

Comment 2:

What are the passive waveguide losses and insertion losses of the couplers and multiplexer in this system?

Response:

The coupling efficiency of the vertical evanescent waveguide coupler was estimated from computations to be approximately 80% of the light impinging from the QCL ridge. This corresponds to an insertion loss of approximately -1 dB. The insertion losses of the multiplexers were simulated and experimentally measured as shown in Fig. 5(d). We estimate them to be -0.5 dB and -1 dB for 6.2-6.3 μm and 7.3-7.4 μm wavelengths, respectively. Finally, the losses for the passive waveguides are estimated to be in the range

1-3 dB/cm based on the measurements that we performed and reported earlier in Ref. ²⁷ for similar waveguides.

To further clarify these points, we added sentences on pp. 4 and 5 in the revised manuscript. We now say on p.4:

“Over 80% coupling efficiency of the power of the forward-propagating QCL mode into the passive waveguide is predicted by simulations at both 6.25 μm and 7.3 μm wavelengths. This corresponds to the coupler insertion loss of less than -1 dB at both wavelengths.”

And on p.5:

“Based on the results shown in in Fig. 5(d), we estimate the insertion losses of the multiplexer at -0.5 dB and -1 dB for 6.2-6.3 μm and 7.3-7.4 μm wavelengths, respectively. We further note that losses for the passive waveguides are estimated to be in the range 1-3 dB/cm for 6.3-7.4 μm wavelengths based on the values measured earlier for a similar waveguiding system ²⁷.”

Comment 3:

The devices were operated in pulsed mode. What constrained them from achieving CW lasing?

Response: This is a direct consequence of the increased threshold current density seen for pulsed operation of the DFB devices in Fig. 7 in comparison to the FP devices in Fig. 4. The supplementary information explains how an accidentally too thickly grown passive core was causing this threshold increase for the narrower ridge-width DFB devices in contrast to the wider FP reference devices.

Comment 4:

The manuscript describes impressive MBE regrowth technology and references previous research using free-space setups for multi-species gas sensing with individual QCL chips. However, it would be valuable to discuss whether hybrid/heterogeneous integration methods, such as butt coupling between different QCL chips and passive waveguides, are feasible. Additionally, a comparison of hybrid/heterogeneous integration versus the monolithic approach in terms of efficiency, scalability, and complexity would strengthen the discussion.

Response: The main advantage of the homogeneous integration technology is that the entire PIC chip is a single crystal which results in high PIC reliability and robustness to environmental changes as discussed, e.g., in Ref. ²⁵. This is particularly important for QCLs that have very high threshold power density of $>10 \text{ kW/cm}^2$, which makes hybrid/heterogeneous integration of these devices difficult, as discussed in Ref. 25. Additionally, monolithic integration of all devices on the same crystal simplifies mass production of PICs based on standard III-V semiconductor processing. We emphasize this point on p.2 of the introduction section in the revised manuscript where we say:

“Here, we experimentally demonstrate the first mid-IR PICs with dissimilar QCL active regions monolithically integrated and optically coupled on the same InP crystal. In this approach, the entire PIC chip is a single crystal which results in its high reliability and robustness to environmental changes. This is particularly important for QCL-based PICs since QCLs have very high threshold power density of over 10 kW/cm^2 ²⁵. Additionally, monolithic integration of all devices on the same crystal simplifies mass production of PICs based on standard III-V semiconductor processing technology.”

Comment 5:

In Figure 3.a, the peak power of the FP QCLs in the grown and regrown regions is around 300 mW and 250 mW, respectively. The difference in performance is less pronounced compared to Figure 7.a for DFB lasers. Does this variation arise from fabrication inconsistencies or fundamental differences between FP and DFB lasers?

Response: Thank you for your question. The two sets of FP lasers in Fig. 3(a) use the same active region design and doping to demonstrate the regrowth capabilities. Devices in Fig. 7 (a) have different active regions and doping. Therefore, a direct comparison of the results shown in Fig. 3(a) and Fig. 7(a) is not possible. Regarding the differences between the performance of FP and DFB devices reported in this work, please see our answer to comment 3 above.

Comment 6:

In Figure 7, DFB lasers with two active regions were characterized, and the power, spectra, and profile were collected from a single passive output waveguide. Were the two lasers pumped and characterized simultaneously, or one at a time? Can the device operate as a multi-wavelength system with both lasers functioning simultaneously?

Response: We had both lasers electrically connected to their respective contact pads but we switched the lasers on one at a time for characterization. However, there are no constraints for operating both lasers simultaneously.

Comment 7:

Defect lines along the interface between grown and regrown sections are evident in Figure 2.f. What is the typical size of these defect lines, and how do they impact device performance? If multiple active regions are regrown to span a broader spectral range, will these defects pose a scalability limit considering device size? Additionally, for broader spectral ranges, will the design of a high-efficiency multiplexer present a challenge in this system?

Response: Thank you for the question. The width of the defect region can be estimated from the SEM image shown in Fig. 4(a). We estimate it to be approximately 10 μm , indicating that a very large number of dissimilar devices may in principle be integrated using the proposed approach. To clarify this point in the revised manuscript, we have added the following on p. 5: “The scanning electron microscope (SEM) image of the cleaved side of the resulting wafer structure is shown in Fig. 4(a). The image allows us to estimate the width of the defect region at the interface between the grown/regrown QCL sections to be $\sim 10 \mu\text{m}$, indicating that very high integration density of dissimilar devices can be attained with this approach in principle.”

The design of high-efficiency multiplexer for more advanced PIC configurations is specific to these particular PIC configurations and is beyond the scope of this report. We note that, in principle, one can consider not only passive multiplexers but also dynamically-controlled (e.g., by voltage or local temperature) switches or multiplexers, depending on the specific PIC requirements.

Comment 8:

The two DFB lasers' wavelengths were designed for NO₂ and SO₂ detection. Has the system been tested for gas sensing applications? Including experimental results or a discussion of this would strengthen the practical relevance of the work.

Response: The main novelty and impact of this report is the new approach to QCL integration. These PICs have been delivered to our partners on this EU project to be tested for gas sensing; however, we do not have the results from them yet. Given that the chips can only be operated in pulsed mode at the moment, we do not expect record-breaking levels of sensitivity, although they are certainly capable of sensing of sufficiently high gas concentrations.

Comment 9:

According to the Methods section, there is an additional etch-stop layer on the active region, but this layer is not shown in Figure 2. Including this in the figure would provide clarity.

Response: Thank you for your suggestion. We added the sacrificial layers in Fig. 2 to depict the discussed regrowth process in full detail. Below you can see the previous version

and the updated version of Fig. 2 shown in the revised manuscript.

We hope that we addressed the questions and comments of the reviewers in sufficient details and we look forward to their further feedback.

Sincerely,

Dominik Burghart and Mikhail Belkin on behalf of all the co-authors

Dear reviewers,

We thank you for carefully reading the manuscript and suggesting improvements.

Please find below our point-by-point response to your questions and comments. We use black font for your comments and blue font for our replies. Additionally, we are submitting an updated manuscript file in which your comments have been addressed. The modified portions of text in the revised manuscript are underlined.

Reviewer #1 and #2 (reviewed the manuscript jointly):

Comment 1:

In comment 6 [of the original review], it is advised to characterize the system with two lasers pumped simultaneously. It could be right that there are no constraints for operating both lasers simultaneously, while it is still necessary to validate the multi-color PIC based on this method. Please provide the spectrum at the output.

Response: We performed the requested measurements. Spectra with each of the two lasers pumped independently and the spectrum of the two lasers pumped simultaneously are now included in the updated Fig. 7(b) shown below. Our measurements show no constraints for operating both lasers simultaneously.

Original (left) and updated (right) panel (b) in Fig. 7 of the manuscript.

The figure caption has been updated to say “(b) Mid-infrared spectra for the lasers in the PIC shown in (a) operated individually (top and bottom panels) as well as simultaneously (middle panel).” The manuscript text has been updated to say “Nevertheless, pulsed room temperature operation was achieved for both devices at the target DFB wavelengths in individual and simultaneous operation as shown in Fig. 7(b).”

Comment 2:

In comment 4 [of the original review], it is advised to give a comparison of hybrid/heterogeneous integration versus the monolithic approach in terms of efficiency, cost, scalability, and complexity. It is true that the monolithic chip is reliable and robust to environmental changes. While it remains to be discussed whether it can simplify mass production in III-V PICs. First, multiple MBE growth and lithography

processes will significantly increase the difficulty of wafer preparation. Second, before we deposit the AR2 layer, half of the AR1 layer was etched away, which will increase the cost. Therefore, a discussion on efficiency and cost will be beneficial.

Response:

In the near-infrared spectral range, state-of-the-art heterogeneously-integrated PICs can achieve ~10 times higher number of photonic elements compared to systems on InP. Nevertheless, both technologies remain commercially competitive, particularly when PICs are designed to have a large number of III-V active element (lasers, modulators, and detectors) and when system reliability is important. Infinera (<https://www.infinera.com/>), for example, manufactures a range of products based on monolithic InP PICs produced by multiple growth/processing/regrowth steps.

Photonic foundries for fabricating both monolithic InP-based (e.g., Smart Photonics, <https://smartphotonics.nl/>) and heterogeneous silicon-based (e.g., IMEC, <https://www.imec-int.com>) PICs are commercially available. Additionally, more InP-based (e.g., Coherent, <https://www.coherent.com/news/blog/indium-phosphide-wafer-fab>) and silicon-based (e.g., TSMC, https://www.photonics.com/Articles/TSMC_Partnerships_Target_Integrated_Photonics/a69934) photonic foundries are being built.

In the mid-infrared spectral range, heterogeneous integration has only been demonstrated for QCLs operating at 4.6-4.8 μm wavelengths and devices could only be operated in pulsed mode with very low duty cycles due to high thermal resistance of heterogeneously-integrated QCLs.

We have tried to summarize these facts into a new paragraph on p. 2 of the introduction section. We now say:

“In the near-infrared spectral range, monolithic integration of III-V lasers with other photonic elements on an InP crystal based on multiple epitaxial growth steps²⁶ as well as heterogeneous integration of III-V lasers with other active elements on silicon-on-insulator (SOI) or other waveguiding platforms²⁷, are used to produce near-infrared PICs containing dissimilar active and passive optical elements. Both approaches result in commercially successful products^{27,28}. However, established near-infrared waveguiding platforms are not transparent across the entire mid-IR spectral range, particularly beyond 5 μm ²⁹. Additionally, QCLs have threshold power densities approximately 10-100 times higher than that of diode lasers^{7,25}, which makes thermal management of heterogeneously-integrated QCLs more difficult than that of diode lasers, which is already a challenge in heterogeneous PICs²⁷. At present, only low-duty-cycle short pulse QCL operation have been achieved for heterogeneously-integrated QCLs and all devices operated at wavelengths shorter than 5 μm ³⁰⁻³². In contrast, the InP-based waveguiding platform is transparent in 3-15 μm ³³⁻³⁵ and monolithic integration of QCLs on InP allows one to follow standard approaches for thermal management developed for InP-based QCLs.”

Comment 3: The whole paper is focused on the fabrication process, there is still significant room for improvement in the device and application. In figure 7 (d), why the LI curve oscillates greatly when J is larger than 4.5 KA/cm². The article provides two possible solutions without implementation. Will the device fulfill its designated function in this scenario? I do not intend to imply that this article lacks novelty. However, further improvement in the device’s performance would enhance the article’s persuasiveness.

Response:

The origin of oscillations in the LI curve in Fig. 7(d) is explained on p. 7 of the revised manuscript. We say “The results are shown in Fig. 7(d). The etalon effect of the passive waveguide section on the DFB QCL laser performance can clearly be observed as regular oscillations in the light-current-voltage characteristics displayed in Fig. 7(d). The period of optical power oscillation in the L-I device characteristic is 0.9 W of electrical power dissipation and it is consistent with the tuning rate of the 6.25 μm DFB lasers of -0.74 cm^{-1} per 1 W of electrical power dissipation and the Fabry-Perot mode spacing of the 2.4-mm-long passive waveguide section of the PIC of $\Delta\nu=1/(2n_{\text{eff}}L)\approx 0.67 \text{ cm}^{-1}$, where $n_{\text{eff}}\approx 3.13$ is the effective refractive index of TM_{00} mode in the passive waveguide. This power oscillation can be suppressed by introducing a nearly-perfect anti-reflection coating on the output facet as shown in the Supplementary Information.”

We have deposited anti-reflection coating on the output waveguide facet of the PIC tested in Fig. 7(d) and experimentally confirmed that the power oscillations are indeed suppressed. Unfortunately, the antireflection-coated PIC required some cooling to achieve continuous-wave operation. Details are provided in the revised Supplementary Information section, where we say:

“2. Suppression of power oscillations during CW operation by anti-reflection coating of the passive waveguide facet

Power output oscillations were observed for the PIC operating in CW regime in Fig. 7(d) of the main text. The oscillations were attributed to the etaloning effect due to light reflection from the output facet of the passive waveguide. A dielectric anti-reflection (AR) coating may be used to suppress waveguide facet reflectivity to avoid the etaloning effect. We have deposited AR coatings on the output waveguide facet of the PIC tested in Fig. 7(d) and indeed observed a complete removal of the etaloning effect in the light output-current characteristics. Unfortunately, CW operation of the PIC with AR coating could only be observed with some cooling as illustrated in Fig. 2S. As shown, the device has a linear slope efficiency over a range of 3.2 W in electrical power consumption, which would result in approximately 3.5 oscillations of the output power for an uncoated device (cf. Fig. 7(d)).

Figure 2S. CW light-output-current characteristics of the PIC tested in Fig. 7 with AR coating deposited on the output facet. The AR-coated device shows a linear slope efficiency for the entire range of 3.2 W of power consumption highlighted by the light grey shaded region. Measurements are performed at 200°K.”

We hope that we addressed the questions and comments of the reviewers in sufficient details and we look forward to their further feedback.

Sincerely,

Dominik Burghart and Mikhail Belkin on behalf of all the co-authors

This manuscript presents a novel approach to monolithic integration of QCLs with distinct active regions on a single InP substrate. The work demonstrates significant advancements in mid-infrared PICs with applications in spectroscopy, gas sensing, and multi-wavelength communication systems. While the results are promising, the manuscript requires substantial revisions to address several technical and contextual gaps.

1. The threshold current density of the devices is relatively high. How does this compare with state-of-the-art QCLs in the literature?
2. What are the passive waveguide losses and insertion losses of the couplers and multiplexer in this system?
3. The devices were operated in pulsed mode. What constrained them from achieving CW lasing?
4. The manuscript describes impressive MBE regrowth technology and references previous research using free-space setups for multi-species gas sensing with individual QCL chips. However, it would be valuable to discuss whether hybrid/heterogeneous integration methods, such as butt coupling between different QCL chips and passive waveguides, are feasible. Additionally, a comparison of hybrid/heterogeneous integration versus the monolithic approach in terms of efficiency, scalability, and complexity would strengthen the discussion.
5. In Figure 3.a, the peak power of the FP QCLs in the grown and regrown regions is around 300 mW and 250 mW, respectively. The difference in performance is less pronounced compared to Figure 7.a for DFB lasers. Does this variation arise from fabrication inconsistencies or fundamental differences between FP and DFB lasers?
6. In Figure 7, DFB lasers with two active regions were characterized, and the power, spectra, and profile were collected from a single passive output waveguide. Were the two lasers pumped and characterized simultaneously, or one at a time? Can the device operate as a multi-wavelength system with both lasers functioning simultaneously?
7. Defect lines along the interface between grown and regrown sections are evident in Figure 2.f. What is the typical size of these defect lines, and how do they impact device performance? If multiple active regions are regrown to span a broader spectral range, will these defects pose a scalability limit considering device size? Additionally, for broader spectral ranges, will the design of a high-efficiency multiplexer present a challenge in this system?
8. The two DFB lasers' wavelengths were designed for NO₂ and SO₂ detection. Has the system been tested for gas sensing applications? Including experimental results or a discussion of this would strengthen the practical relevance of the work.
9. According to the Methods section, there is an additional etch-stop layer on the active region, but this layer is not shown in Figure 2. Including this in the figure would provide clarity.